# Stochastic Dynamic Mass Spectrometric Quantitative and Structural Analyses of Pharmaceutics and Biocides in Biota and Sewage Sludge

**DOI:** 10.3390/ijms24076306

**Published:** 2023-03-27

**Authors:** Bojidarka Ivanova

**Affiliations:** Independent Researcher, 44139 Dortmund, Germany; bojidarka.ivanova@yahoo.com

**Keywords:** mass spectrometry, stochastic dynamics, surfactants, biota, sludge, quantitative and 3D structural analyses

## Abstract

Mass spectrometric innovations in analytical instrumentation tend to be accompanied by the development of a data-processing methodology, expecting to gain molecular-level insights into real-life objects. Qualitative and semi-quantitative methods have been replaced routinely by precise, accurate, selective, and sensitive quantitative ones. Currently, mass spectrometric 3D molecular structural methods are attractive. As an attempt to establish a reliable link between quantitative and 3D structural analyses, there has been developed an innovative formula [
$D_{SD}^{\prime \prime ,tot}=\sum_{i}^{n}D_{SD}^{\prime \prime ,i}=\sum_{i}^{n}2.6388.10^{-17}\times \bar{I_{i}^{2}}-{\left(\bar{I_{i}}\right)}^{2}$
] capable of the exact determination of the analyte amount and its 3D structure. It processed, herein, ultra-high resolution mass spectrometric variables of paracetamol, atenolol, propranolol, and benzalkonium chlorides in biota, using mussel tissue and sewage sludge. Quantum chemistry and chemometrics were also used. Results: Data on mixtures of antibiotics and surfactants in biota and the linear dynamic range of concentrations 2–80 ng.(mL)^−1^ and collision energy CE = 5–60 V are provided. Quantitative analysis of surfactants in biota via calibration equation ln[*D*″_SD_] = *f*(conc.) yields the exact parameter |r| = 0.9999_1_, examining the peaks of BAC-C12 at *m*/*z* 212.209 ± 0.1 and 211.75 ± 0.15 for tautomers of fragmentation ions. Exact parameter |r| = 1 has been obtained, correlating the theory and experiments in determining the 3D molecular structures of ions of paracetamol at *m*/*z* 152, 158, 174, 301, and 325 in biota.

## 1. Introduction

Biocides and antibiotics are major classes of chemicals used to control or prevent the growth of microorganisms such as fungi, mosses, bacteria, lichens, and algae [1,2,3]. The therapeutics are efficacious at low concentration levels, due to specific interactions with single cellular targets. Conversely, biocides adsorb on a microbe’s surface or are utilized for suspension at higher concentration levels compared with their minimum inhibitory concentrations. Their monitoring in the aquatic environment is of primary and increasing concern as well [4]. The main biocidal disposal route is via sewage systems and drains. The potential risk for bioaccumulation carries toxicological effects on the ecosystem [1,5,6,7,8]. The major chemical compositions of cleaning substances, including disinfectants, cosmetics, and personal care products, are surfactants [9,10,11,12]. Surfactants can be found in paints, polymer materials, fabrics, pesticides, and pharmaceutical products, in addition to oil, mining, and cellulose factories. They are emerging contaminants according to UNESCO [13]. The monitoring of the environmental and wastewater pollution of surfactants is a primary research task. Quaternary ammonium surfactants or so-called quats are toxic to aquatic organisms at a level of concentration of 1 mg.L^−1^ [1,14,15]. However, quaternary ammonium surfactants are not only toxic to environmental organisms, but also exhibit the proliferation of antibiotic resistance [16]. In 2019, there were 4.95 million antibiotic-resistant pathogens around the world, which has been considered as a silent pandemic [10]. The therapeutics’ abuse is a factor contributing to antibiotic resistance. It remains not well understood, but is recognized to be among the most pressing global environmental and human health problems [17,18]. It has been found that by 2050, approximately ten million people shall die, annually, as a result of antibiotic resistance [19]. Despite this, many quats play a vital role in biochemical pathways [20]. Benzalkonium chlorides (Figure 1) belong to the group of quaternary ammonium surfactants, and are extensively utilized for various industrial and domestic purposes [10,21,22], including contact lens solutions or eye drops [23,24,25]. For this reason, BAC-C14 is the most frequently detected biocide among benzalkonium chlorides in engineered and environmental systems [16]. Their presence in wastewater upset has led to activated sludge processes [26,27]. Benzalkonium chloride derivatives in sludge are capable of biodegrading during biological wastewater treatment or adsorbing onto a biomass. Their biodegradation paths have been examined [28,29,30]. The presence of benzalkonium chloride disinfectants in the environment promotes the abundance and diversity of antibiotic resistance genes in sewage sludge microbiomes [10]. Antibiotics, as surfactants and environmental pollutants, represent a widespread concern and ecotoxicological risk [23,24,25,31,32].

Atenolol and propranolol are *β*-blockers treating cardiovascular diseases [33]. They are emerging pollutants, found in sewage effluents and surface water. *Beta*-adrenergic receptors, which are major target macromolecules of *beta*-blockers, were detected in aquatic animals and fish. There is an environmental effect of the pharmaceutics on physiological processes in wild animals. The same can be said for paracetamol (tylenol or acetaminophen), which is a commonly used anti-inflammatory, antipyretic, and analgesic medication [32,34,35]. It can be found in tap water from the pharmaceutical industry and in urban and hospital waste [36]. There have been developed innovative strategies for the regulation of pharmaceutics in surface and groundwater via European legislation (Directives 2000/60/EC, 2008/105/EC, 2009/90/EC, 2013/39/EU, and 2015/1787/EU [37]).

The determination of pharmaceutics and biocides in biota, including the analysis of invertebrates, plankton, fish, human tissues and fluids, birds, and marine mammals, is importance for assessing the risk for human health and environmental damage [38,39], as well as determining the potential *ecological risk index* [38,39,40]. The *biomonitoring* includes saltwater and freshwater mussels [1,41,42,43]. These biological species filter large quantities of water, thus accumulating both organic and inorganic pollutants from water or suspended matter at elevated levels. In addition, mussels are (i) widely distributed in the environment; (ii) sessile biological species; (iii) thrive in highly polluted areas; and (iv) easily sampled [41]. The biomonitoring of pollutants via mussels allows us (a) to determine the concentration levels of pollutants in feral organisms; (b) to examine the spatial (geographic) distribution of pollution, if any; and (c) to study the temporal distribution and variation of pollution toward distinct seasons. The *toxicokinetics* examines four major processes of biological objects such as mussels, involving (a) the adsorption of pollutants; (b) their distribution, (c) metabolism, and (d) excretion [44]. The rates of the processes determine the *contaminant’s bioavailability*. Knowledge of the adsorption capability of pollutants is of importance, because the process is also used to decontaminate water. It plays a crucial role in methods for the removal of organics in wastewater treatment plants, thus highlighting the aerobic and anaerobic sludge systems [45,46]. The importance of the development of methods for determining surfactants and antibiotics in mixtures in biological and environmental samples is due to the fact that benzalkonium chlorides, for example, are used in commercial pharmaceutical formulations [47]. Catanionic mixtures of drug-surfactant aggregates such as benzalconium chlorides and β-blockers such as alprenolol, ATE, and PRO have been detailed [21,22]. β-Blockers have been used to treat ocular hypertension and glaucoma. Their pharmaceutical formulations with benzalconium chlorides have been examined in skin creams or eye drops [48].

Routinely, MS methods have been used to determine organic pollutants [1,49,50]. The ultra-high resolving power, selectivity, accuracy, precision, and sensitivity of tandem mass spectrometry are irreplaceable, being used for the analysis of environmental and biological samples. However, MS cannot be used as a universal method for determining mixtures of quaternary ammonium derivatives simultaneously [5,6,27,51].

*Analytical mass spectrometry* is a complex term, referring to three major research tasks of *qualitative*, *quantitative*, and *structural analyses*. However, pursuing the exact *chemometrics* (|r| = 1) is a challenging analytical task [52,53,54,55,56,57,58]. Among theoretical model equations that are particularly relevant are those producing exact method performance. Accordingly, research effort has been devoted to developing MS methods for data processing, ensuring the quality and comparability of analytical information toward the interpretation of results [59], which is in agreement with Council Directives 96/23/EC and 2002/657/EC.

However, depending on the analyte concentration in biological, foodstuff, and environmental samples, there is observed a decrease in method performance using classical methods for the data processing of MS measurands. How do we address this problem? It has been considered via innovative stochastic dynamic Equations (1) and (2), capable of exact quantifying analytes [55,56,60,61,62,63,64,65,66,67]. Formula (2) is derived from Equation (1), where ‘I’ denotes the intensity of the MS peak. There are excluded imaging methods for quantifying analytes or approaches capable of spatially resolving the chemical composition of a surface sample [68].

Formula (2) overcomes a set of difficult classical quantitative MS concepts [55,56,60,61,62,63,64,65,66,67]. Superior chemometrics is explained with the help of Formula (2) to quantify exactly the fluctuations in measurands in a short period of scan time. It is capable of determining the 3D molecular and electronic structures of chemicals mass-spectrometrically, when it is used complementarily with Arrhenius’s Equation (3). The functionality *D*″*_SD_* = *f*(*D*_QC_) has shown |r| = 0.9999_4_ [65].

To summarize, the study deals with the quantitative and 3D structural stochastic dynamic MS biomonitoring of mixtures of antibiotics PARA, ATE, and PRO in the presence of benzalkonium chloride surfactants BAC-C12, BAC-C14, BAC-C16, and BAC-C18 in biota using mussel tissue, sludge cakes, and a treated effluent.

(1)
$$D_{SD}^{tot}=\sum_{i}^{n}D_{SD}^{i}=\sum_{i}^{n}1.3194.10^{-17}\times A^{i}\times \frac{\bar{I_{i}^{2}}-{\left(\bar{I_{i}}\right)}^{2}}{\bar{{\left(I_{i}-\bar{I_{i}}\right)}^{2}}}$$


(2)
$$D_{SD}^{\prime \prime ,tot}=\sum_{i}^{n}D_{SD}^{\prime \prime ,i}=\sum_{i}^{n}2.6388.10^{-17}\times \bar{I_{i}^{2}}-{\left(\bar{I_{i}}\right)}^{2}$$


(3)
$$D_{QC}=\frac{\prod_{i=1}^{3N}\nu_{i}^{0}}{\prod_{i=1}^{3N-1}\nu_{i}^{s}}\times e^{-\frac{\Delta H^{\#}}{R\times T}}$$


## 2. Results and Discussion

### 2.1. Mass Spectrometric Data

#### 2.1.1. Mass Spectrometric Fragmentation Reactions of Paracetamol and Its d_3_-Derivative

Paracetamol shows fragmentation path CID (*m*/*z* 152)→152, (134,) 110 (Figure 2, Appendix A) [69,70,71,72,73,74,75,76,77,78,79] (see chemical diagrams of species in Appendix A). Its dimer [2M+H]^+^ shows fragmentation path CID (*m*/*z* 303)→303, 152 [69,75]. The ammonium adduct [2M+NH_4_]^+^ exhibits a low-abundance peak at *m*/*z* 320 [75]. MS analysis of PARA radical-cations has been reported [72]. Fragmentation processes, involving a radical-cation mechanism of bond cleavage, have been proposed [77]. PARA tends to stabilize not only the Cu^2+^ adduct, but also adducts of alkali metal ions and NH_4_^+^ cation. There are species of type [M+NH_4_]^+^ (*m*/*z* 169), [M+Na]^+^ (*m*/*z* 174), [M+K]^+^ (*m*/*z* 190), [2M+NH_4_]^+^ (*m*/*z* 320), [2M+Na]^+^ (*m*/*z* 325), and [2M+K]^+^ (*m*/*z* 341), respectively [79]. As Appendix A reveal, the abundance of peaks depends on the applied voltage, presence of formic acid, and analyte concentration. The same is true for the peak of protonated analyte [M+H]^+^ and its major fragmentation product of N–C bond cleavage, [M-CH_3_CHC=O]^+^, at *m*/*z* 152 and 110. The data on d_3_-PARA show similar fragmentation patterns together with some adducts. The peak at *m*/*z* 331 belongs to [2.d_3_-M+Na]^+^. There is an observed fragmentation reaction CID (*m*/*z* 152)→152, 110, 93 [78]. MS spectra of PARA in a sludge cake and biota (Figure 3, Appendix A) show competitive fragmentation mechanisms causing not only charged cations, but also cation radicals.

#### 2.1.2. Mass Spectrometric Fragmentation Reactions of Biocides

Surfactants BAC-C12, BAC-C14, BAC-C16, and BAC-C18 exhibit molecular cation [M]^+^. The major fragmentation path shows the loss of 92 Da of toluene (Appendix A). Regarding the MS spectra of PARA and its d_3_-PARA derivative (Appendix A), depending on the experimental conditions, there are competitive fragmentation reactions causing more than one conformational and tautomeric form of product ions (Appendix A). Although the major fragmentation MS path of these surfactants is associated with the loss of the hydrophilic head of the compounds, the product ion consisting of a charged hydrophobic tail exhibits a complex conformational preference and electronic effects (Appendix A). Therefore, after examining only experimental measurands in CID-MS/MS and SRM operation modes, a lack of assignment of observable peaks to ions remains. Consider the shape of the SRM spectrum of BAC-C12 (Appendix A) and the proposed chemical 2D diagrams and electronic structures of ions at *m*/*z* 212 and 213.

#### 2.1.3. Mass Spectrometric Fragmentation Reactions of β-Blockers

Propranolol shows the [M+H]^+^ cation at *m*/*z* 260 [80]. The fragmentation paths depend on CE, pH, etc. [1,80,81,82,83,84,85,86,87,88,89,90,91,92,93,94,95]. With increasing CE, there is a low-abundance ion at *m*/*z* 183, due to the cleavage of the [C_3_H_9_N]^0^ fragment and solvent water [1,80,83] (Appendix A). The peak at *m*/*z* 282 of the MS spectrum of PRO at CE = 30V is assigned to the [M+Na]^+^ adduct. The same is true for ATE MS reactions (Appendix A). Their identical structural (2,3-dihydroxy-propyl)-isopropyl-ammonium fragment causes peaks at *m*/*z* 145, 105, 101, 83, and 64, respectively. ATE exhibits the [M+H]^+^ cation at *m*/*z* 267. Quantitative analysis was carried out, examining the [M+H]^+^ cation at *m*/*z* 260 and 267 of PRO and ATE in the mixture. Employment of the SRM and SIM modes leads to pairs of MS peaks at 260/261 and 267/268 (Appendix A). Classical quantitative methods look at average *m*/*z* data on MS peaks at *m*/*z* 260.5 (SRM) and 262.07 (SIM) (PRO), as well as 267.52 (SRM) and 267.82 (SIM) (ATE). The matrix affects significantly not only the *m*/*z* data compared with the results from the fragmentation paths of standard samples, but also product ions [86,87,88,89,90,91,92,93,94,95] (Appendix A).

Owing to the fact that there is used only an MS peak of the [M+H]^+^ cation in both the cases, excluding a statistically representative set of fragmentation species, there is particular importance in accounting for the molecular conformations and electronic effects of protonated antibiotics in order to assign statistically different sets of *m*/*z* data on two SRM and SIM spectra of standard samples of antibiotics and their mixtures in environmental and biological matrixes. For the purposes of 3D structural analysis and empirical demonstration of the assignment of the [M+H]^+^ cation at *m*/*z* 260 of PRO in the complex sample matrix, there is used a statistically representative set of MS peaks at *m*/*z* 260, 283, 157, 116, and 98 found in the MS spectrum in soil [80]. The same fragmentation species have been found in the MS spectrum of PRO in biological samples [81,91]. Owing to the proposed two competitive mechanisms of formation of the MS ion at *m*/*z* 183, our study examines the correlation between MS data and theoretical quantum chemical ones looking at ions 183__a_ [80,81,90] and 183__b_ [82]. The MS ion at *m*/*z* 116 in the CID-MS/MS spectrum of PRO has been observed examining tandem MS/MS processes of ATE and studying the ions’ CID interaction of the [M+H]^+^ cation at *m*/*z* 267 [84]. The peak at *m*/*z* 116 has been used to determine PRO in the liver, brain, and kidney thin tissue, as well [85,94]. In addition to the peak at *m*/*z* 116, ATE exhibits a set of ions depending on the experimental conditions [96,97,98,99,100]. These are peaks at *m*/*z* 225, 208, 190, 173, 162, and 145 [84,96,97,98,99,100]. Moreover, there are peaks at *m*/*z* 133, 115, and 107 [94] (Appendix A).

### 2.2. Determination of Stochastic Dynamic Diffusion Parameters

The capability of Equation (1) in determining the 3D molecular structures of analytes, when it is used complementarily with Equation (3), has already been reviewed [60]. In this light, herein, we prove its validity and compatibility with Equation (2). In verifying empirically the validity of Equation (1), we can explore the results from SRM data on the MS ion at *m*/*z* 110 of PARA of its MS/MS spectra of the [M+H]^+^ cation at *m*/*z* 152 (Appendix A). The new data on variables of PARA show lnP1 = 17.0532. Therefore, Equation (1) shows that the MS law is valid for the temporal distribution of measurands of PARA, as well. Details of the statistical parameters A^i^ are presented in Appendix A. Calculation tasks have been discussed previously [55,56,60,61,62,63,64,65,66,67]. The latter figure illustrates the relation between the *D*′*_SD_* and *D″_SD_* parameters, showing |r| = 0.9995_3_. The deviation from |r| = 1 is a result of the error contribution of the data processing of the temporal distribution of intensity with respect to the scan time or function (I–<I>)^2^ = f(t) fitted to the SineSqr function, thus producing statistical parameter A^i^ (Appendix A).

### 2.3. Quantitative Data on Biocides

We shall support our method by highlighting how the *D″*_SD_ parameters are determined per span of scan time. We shall justify the view that exact relations are obtained when there are quantified fluctuations of measurands with a short span of scan time. The question that we need to address is “Which criteria determine a set of MS measurands with respect to a concrete span of scan time as the true one?”, or which methods are used in order to validate the parameters of Equation (2). In doing so, we use data on the selected reaction monitoring mode of the PARA [M+H]^+^ ion at *m*/*z* 152 of segments of the MS method, where segment (i) has collected a full mass scan set of variables. There are examined segment raw data QC_High_SRM_SEG_CE40_i.raw (i = 1–3) [101,102] (Appendix A). Appendix A lists the output, only, of the fragmentation ion at *m*/*z* 110 for the SRM operation mode, while Appendix A list results from the CID-MS/MS spectra of PARA of its [M+H]^+^ cation at *m*/*z* 152 depending on experimental conditions such as CE and the presence of formic acid. Chemometrics of the normality Shapiro–Wilk test together with ANOVA data in Appendix A are summarized in Appendix A. Data on quality control standard samples of a mixture of antibiotics QC_H_SRM_CE40_3 (segment 3, Appendix A) reveal three groups of *m*/*z* parameters that are mutually significantly different from the perspective of chemometrics (Appendix A). These are a subset of variables of segment 3, shown as QC_High_SRM_SEG_CE40_3_1, QC_High_SRM_SEG_CE40_3_2, and QC_High_SRM_SEG_CE40_3_3. The same is true for the recorded two sets of *m*/*z* variables of the same ion of segment (2) (QC_High_SRM_SEG_CE40_2_1 and QC_High_SRM_SEG_CE40_2_2). The chemometric analysis of datasets of measurands at *m*/*z* 110.1, i.e., QC_High_SRM_SEG_CE40_3_3, QC_High_SRM_SEG_CE40_2_1, and QC_High_SRM_SEG_CE40_1, is statistically equal. In other words, correlative analysis and determination of the *D″*_SD_ parameters of Equation (2) within the framework of three segments of MS spectra is carried out using those sets of measurands that are statistically significantly equal, or those with values at *m*/*z* 110.065. Therefore, there are distinguished quantitatively three sets of peaks at *m*/*z* 110: 110.84_49_ ± 0.05985, 110.06_525_ ± 0.04709, and 110.23885 ± 0.04898. The number of the subset of variables is not extensive when looking at whole datasets of average values over the whole period of measurement. Results from sector (3) (QC_High_SRM_SEG_CE40_3) show a value at *m*/*z* 110.0673. In other words, our approach does not ignore measurable sets of low-abundance variables and their fluctuations.

The examples could be multiplied, in fact, without limit when looking at the dataset of measurands in environmental and biological samples of antibiotics and biocides. The results provide compelling evidence for the advantages of our method. The analysis of the effluent, treated sludge cake, and biota show that PARA and its d_3_-derivative in the presence of antibiotics or biocides reveal peaks at a range of *m*/*z* 107–115, which further complicate not only the qualitative assignment of products, but also the quantification of those single analytes in mixtures, as well as the deuterium exchange processes of d_3_-PARA, if any.

Quantitative data on biocides in biota show r^2^ = 0.9309–0.985 (Appendix A) using the classical approach to quantify the average total intensity of MS peaks over the whole period of measurements [1,103]. The results were data-processed via the ICIS algorithm of peak detection. The Savitzky–Golay smoothing function with baseline correction was used. The ICIS algorithm involves a trapezoidal integration approach [1,5,6,7,104]. In order to account for the effect of the smoothing function on chemometrics, herein, we examined the same relationships, but applying baseline correction and TIA (Figure 4 and Appendix A). Appendix A illustrates relationships among the concentrations of BAC-C12 and BAC-C14 in biota showing |r|= 0.9874_5_ and 0.9922_3_. Despite this, the ICSI or TIA methods show low |r| parameters and high sd(yEr±) for mean values. The linear equation obtained via the direct application of TIA of BAC-C14 is y = −16.1_3779_ ± 14.6_133_ + 6.5_4102_ ± 0.3_3481_.x. The error contribution of the data-processing algorithm affects the main value of the intercept and slope of linear regression, respectively, and the correlation equations. The reliability of quantitative data is complicated when examining biota. There are competitive fragmentation reactions producing both mono-cations and cation radicals. The CID-MS spectra of PARA of freeze-dried biota show peaks at *m*/*z* 108.07 and 109.07. The wet sample shows MS ions at *m*/*z* 107.07, 110.07, and 111.1 (Figure 1, Figure 3 and Figure 4 and Appendix A). The mixtures of biocides show pairs of ions at *m*/*z* 211 and 212 (BAC-C12) and 239 and 240 (BAC-C14), instead of a single ion according to the common fragmentation scheme. The data processing of isotopologies in biota via the latter algorithms yields high sd(yEr±) values (Appendix A). Either by the ICIS or TIA algorithm, with or without baseline correction, the uncertainty of the analytical results is increased.

The task can be completed precisely by Equation (2). ICIS and TIA are incapable of providing reliable analytical data on environmental and biological samples, particularly when there are tautomers and fragmentation reactions involving different molecular-level mechanisms, leading not only to cations, but also to cation radicals. Moreover, it is true that ca. 25% of pharmaceutics exist in more than one tautomeric form, in addition to the fact that almost all antibiotics are characterized by multiple ionizable protonation positions in their molecules [105]. Reference [63] concerns the same problem of quantifying LMW antibiotics in biological fluid.

Thus, the following paragraph focuses on the quantitative analysis of biocides using Equation (2). We shall describe the advantages of Formula (2) compared with the results from the ICIS or TIA methods presented so far.

Appendix A summarizes *m*/*z* data on BAC-C12 and BAC-C14 in biota at concentrations c = 2–80 ng.mL^–1^. ANOVA and *t*-tests show (Appendix A) two sets of *m*/*z* values at 212.2_2211_ ± 0.12988 and *m*/*z* 211.6 at c = 2 ng.mL^−1^ (BAC-C12), which are statistically significantly different. At c = 6 ng.mL^−1^ are distinguished three sets of measurands at *m*/*z* 213, 212, and 211.5 (Appendix A, Appendix A). There are three elemental compositions, molecular conformations, and electronic structures of BAC-C12 species (212.2_1401_ ± 0.1001, 212.5_4356_ ± 0.12232, and 211.7_86_ ± 0.15203.) Datasets at *m*/*z* 212.22_211_ ± 0.12988 and 212.2_1401_ ± 0.1001 of BAC-C12 ions at c = 2 and 6 ng.mL^−1^ are statistically not significantly different. The ion at *m*/*z* 212.2 belongs to one and the same ion at two concentrations.

Further, we shall come to see that Equation (2) accounts precisely for the fluctuations in the *m*/*z* and intensity data on MS peaks, thus producing excellent-to-exact quantification and 3D structural analysis, despite the complexity of the isotope shape. Figure 5 and Figure 6 and Appendix A show that the quantification of biocide BAC-C12 in biota employing the *D″*_SD_ parameter and assessing the relationship ln[*D″*_SD_] = f(conc.) yields |r| = 0.9999_1_–0.9905_8_, examining c = 2–80 ng.(mL)^−1^. Conversely, the ICIS and TIA algorithms produce |r|= 0.98924 (Appendix A). There has been obtained |r| = 0.999 when studying the same set of analytes in sludge [5]. Data on PARA and d_3_-PARA c = 5–400 ng.(mL)^–1^ show r^2^ = 0.997. Quantification of BAC-C12 and BAC-C14 yields r^2^ = 0.987 and 0.983 [1,6,7]. The analysis of the peak at *m*/*z* 211.75 ± 0.15 of BAC-C12 ions in biota using the [M+H]^+^ cation at *m*/*z* 304 (Appendix A and Figure 6) produces |r| = 0.9972_1_ and 0.9841_1_ when employing the equation *D″*_SD_ = f(conc.) Again, there is improved method performance. The analysis of BAC-C14 yields |r| = 0.9918_8_ within concentration range c = 20–80 ng.(mL)^−1^.

### 2.4. Quantitative Functions between Mass Spectrometric Stochastic Dynamic Diffusion Parameters and Theoretical Total Intensity Variables with Respect to Experimental Parameter Collision Energy

Since the main goal of the current paper is to advocate for a general innovative approach to quantifying analytes in complex environmental and biological matrixes, mass-spectrometrically via Equation (2), in this short subsection, we shall direct the reader’s attention to Equation (4), appearing valid for MS data on labetalol [64].

(4)
$$\bar{I^{TOT,q}}=\frac{1}{2}\times \frac{A_{I}^{q}}{A_{D}^{q}}\times D_{SD}^{\prime \prime ,tot}$$


Equation (4) is derived from Equation (1) (see Equation (A6) in [64]). It connects the theoretical average intensity data on analyte MS ions obtained toward CE and the *D″*_SD_ of Equation (2). Statistical parameters A_D_^q^ and A_I_^q^ are functional amplitudes of the SineSqr function fitted with relation *D″*_SD_^q^ = f(CE) and <I>^q^ = f(CE) of q^th^ MS fragment ion. We look at new empirical proof of the validity of Equation (4). It has been found by examining PARA measurands (Appendix A). Appendix A depict the relations of *D″*_SD_^q^ = f(CE) and <I>^q^ = f(CE) of MS ions of PARA at *m*/*z* 60 and 64. The theoretical <I>_theor_ data on ions with respect to CEs and A_D_^q^, A_I_^q^ parameters (Appendix A) correlate with the experimental <I>_exp_ ones and show |r|= 0.9942_9_ and 0.96501. Since, so far, we have considered only two cases of the application of Equation (4) for the latter purposes, we are unable, currently, to assess the apparent violation of its validity.

### 2.5. Theoretical Data

#### 2.5.1. 3D Molecular Conformations and Electronic Structures of Analytes and Energetics

The calculation of the D_QC_ parameters of Equation (3) has been discussed [61,63,65,66]. However, we need to discuss the correlation between the 3D molecular conformation of MS ions and their energetics, thus highlighting the advantages of Equation (3), consisting of significant sensitivity and selectivity and capable of distinguishing quantitatively among molecular structures, exhibiting subtle electronic effects (Appendix A). Appendix A detail the static and MD DFT results from ions in GS and TS states. Appendix A summarizes the atomic coordinates of fragmentation ions, allowing us to extract geometry parameters such as bond lengths and angles. The energy difference in the fragmentation species of surfactants such as ions 212__a_ and 212__b_ is ∆E^TOT^ = |0.015| a.u. The difference in the energetics of molecular ion [M+H]^+^ of PARA and d_3_-PARA is of the same magnitude order (∆E^TOT^ = |0.01| a.u.). In these and many more cases of ions [60,61,63,65,66], there is provided ample proof favoring Equation (3) as a sensitive and selective tool, allowing us to distinguish among molecular species exhibiting comparable energetics. There are almost identical ∆E^TOT^ ions of tautomers of PARA and d_3_-PARA (Appendix A). The examples of species provide us with real insights into the complexity of the electronic effects and dynamics of MS ions, which are unable to be tackled precisely when examining only free Gibbs energy data on the global minimum of PES (Appendix A). Despite the fact that there is ∆E^TOT^ = |0.01| a.u. of ions 152__a_ and 155__a_ for the molecular cation of PARA and d_3_-PARA, the difference in the D_QC_ parameter is ∆D_QC_ = |3.371| (Table 1).

#### 2.5.2. Determination of Quantum Chemical Diffusion Data

Details on the calculation tasks of the D_QC_ parameters of Equation (3) can be found in [62]. Methodologically, we use vibrational data on MS ions at GSs and TSs. Variations and changes in the energetics of species can be examined adequately via Born–Oppenheimer MD. Table 1 summarizes the D_QC_ parameters of the studied herein MS ions.

### 2.6. Correlative Data on Mass Spectrometry and Quantum Chemistry

We return to the major question that we posed at the beginning of the study: How does Equation (2) serve as a tool to determine the 3D molecular and electronic structures of analytes mass-spectrometrically, even examining multicomponent environmental and biological samples, having complex sample matrix effects? In the latter response, we shall focus on the chemometrics of relation *D″*_SD_ = f(D_QC_). In line with our previous studies devoted to the same issue [61,63,65,66], achieving such a goal requires the assessment of the statistical significance of the mutual relationship between D_QC_ and *D″*_SD_ data on ions belonging to one and the same molecular structure. Figure 7 shows the chemometric results from PARA at *m*/*z* 152, 158, 174, 301, and 325 depending on CE (Table 1 and Appendix A). There are |r| = 0.9979_8_ at CE = 10 V and |r| = 1–0.9936_1_ at CE = 25 V.

Further, little contract might be observed in Appendix A, depicting data on d_3_-PARA and propranolol (Table 1, Appendix A). In the former case, there is obtained |r| = 0.9993_1_. Relation *D″*_SD_ = f(*D_QC_*) of PRO ions at *m*/*z* 260, 157, and 116 shows |r| = 0.9916_1_.

## 3. Discussion

Since the purpose of the study is to gain insights into quantitative functionalities among MS measurands of analytes’ molecular and fragmentation peaks, the physico-chemical properties and parameters of molecular and ionic species, and their 3D molecular and electronic structures, as well as experimental factors and parameters of measurements, this section might be regarded as room for debate, for which we suggest that the discussion helps the reader to understand whether Equations (1) and (2) are capable of providing not only the exact quantification of analytes in complex biological and environmental matrixes, but also the simultaneous 3D structural determination of the same compounds and samples. However, before embarking on a discussion of the advantages of Equations (1) and (2), we provide a few remarks on the data reported so far.

To begin with, methodological contributions devoted to developing quantitative methods for the analysis of datasets of measurands and those devoted to elaborating methods for 3D structural MS analysis are not equally frequent. Therefore, developed methods for simultaneous quantitative and 3D structural analyses, and approaches capable of providing the exact determination of the amounts and structural parameters of molecules, are restricted. We draw the reader’s attention to the fact that Equation (2) is one of the scarce examples of formulas used for both quantitative and 3D structural analyses. However, the latter statements lead us to a logical question: Why should we be forced to become aware of details of analytes’ 3D molecular structures, owing to the fact that quantitative analytical mass spectrometry represents different areas of structural mass spectrometry? Routinely, we process MS-based quantification as a separate research task. It is well known that with such distinctions in research tasks, we are completely able to characterize and quantify analytes mass-spectrometrically. This combined set of research tasks would seem to complicate the further experimental design of the MS analysis of environmental and biological samples. An answer to such a question, if any, would be that, since stochastic dynamic model Equation (2) is a novel analytical MS law, it would be best to provide, herein, an immediate illustration of the crucial importance of the capability of Formula (2) in quantifying and determining the 3D structures of analytes via MS for the purpose of the quantitative analysis of complex environmental and biological samples. Our earlier and most recent outcome of the application of Equation (2) to determine exactly LMW analytes in biological fluids [63]—which, however, largely matches the results from the current study—perhaps best illustrates the advantages of our stochastic dynamic theorization of MS phenomena via Equation (2) over classical quantitative approaches. For instance, these include the ICIS or TIA algorithms, dealing with the integration of the area of the MS shape of analyte fragment peaks as a continuous function of the *m*/*z* values with respect to MS intensity, instead of as discrete random variables and their fluctuations with a short span of scan time, as according to Formulas (1) and (2). As the MS analysis of metronidazole in clinical human urine has demonstrated [63], analyte molecular ion [M+H]^+^ is characterized by a set of statistically significantly different *m*/*z* variables depending on the experimental conditions, but explicitly highlighting low analyte concentrations at a range of 2.5 to 25,000 ng.(mL)^−1^. It has been found that there are observed mass-spectrometrically two datasets of measurands at *m*/*z* 172.071_8_ and 172.040_81_ of the [M+H]^+^ cation. As can be expected, the employment of classical quantitative approaches to determine the analyte concentration yields |r| = 0.9939_5_–0.9940_4_ using linear calibration equation I^TOT^ = f(conc.), where I^TOT^ is determined via the ICIS or TIA algorithms. The decrease in method performance has been explained with the fact that, on the one hand, there are quantified isotope shapes of two different *m*/*z* quantities, which even can belong to two different analytes in complex biological samples, when there are determined unknown compounds. On the other hand, the error contribution to the mathematical data processing of MS patterns by means of the ICIS or TIA algorithms is significant, due to the large sd(yEr±) values of integration approaches. Due to these reasons, we suggest that the reliable and exact quantitative analysis of such complicated cases of fluctuations of MS measurands at very low analyte concentrations and complex matrix effects can be carried out exactly, accurately, precisely, selectively, and sensitively only via Equation (2) and the simultaneous quantitative and 3D structural analysis of analytes with respect to the experimental conditions of measurements. These combined research tasks allow us to assign exactly statistically different sets of measurable variables to corresponding molecular conformations and electronic structures of analytes. It has been found that the aforementioned MS peaks of metronidazole at *m*/*z* 172.071_8_ and 172.040_81_ belong to its two different tautomeric forms. The performed quantitative analysis based on two different calibration statistical equations *D″*_SD_ = f(conc.) of two fragmentation peaks has resulted in |r| = 1. Turning to the results from this study in quantifying biocides BAC-C12 and BAC-C14 in biota, it can be easily shown an analogous cases of the temporal distribution and variations of MS measurands of these analytes in complex matrix samples depending on the experimental conditions, particularly highlighting a low analyte concentration as a major factor causing the observation of sets of statistically different *m*/*z* measurable variables belonging to different molecular conformations and electronic structures of analyte fragmentation species. Due to these reasons, the employment of the ICIS or TIA algorithms, quantifying the isotope shape area of function *m*/*z* = f(I), yields |r| = 0.9304–0.9856. Conversely, as the results from our analysis using Equation (2) show, there are statistically significant sets of variables of fragmentation ions at *m*/*z* 212 of CAB-C12 obtained as a result of the SRM tandem fragmentation mode of the molecular cation of analyte [M+H]^+^ at *m*/*z* 304 (Appendix A). The statistical linear calibration models ln*D″*_SD_ = f(conc.) and *D″*_SD_ = f(conc.) have resulted in exact method performance, showing |r| = 0.9999_1_–0.9905_9_ and 0.9972_1_. There are examined MS peaks of BAC-C12 at *m*/*z* 212.209 ± 0.1 and 211.75 ± 0.15. Thus, again, there is observed a highly reliable and very prominent quantitative analysis when we use Equation (2) instead of classical quantitative MS approaches based on the aforementioned algorithms. The new data presented in this paper on the MS quantitative analysis of mixtures of biocides and antibiotics in biota and sewage sludge show clearly the capability of the exact and reliable processing of complex isotope shapes of MS measurands, obtained as a result of competitive processes of tautomers and mechanisms involving the formation of cations and cation radicals, which is not only not universal, but also is beyond the capability of classical quantitative methods for the data processing of MS measurands. It is reasonable to assume, therefore, that classical automated algorithms of data processing of observable variables of such complex MS patterns are of little use when dealing quantitatively with the analysis of environmental and biological multicomponent samples of unknown analytes, very low analyte concentrations, and sample matrix effects. Furthermore, owing to the fact that the determination of analytes within the framework of the stochastic dynamic theory and model Equation (2) is carried out without the presence of IS, it is obvious that the innovative method allows researchers to determine quantitatively and structurally by mass spectrometry any unknown analyte in a complex mixture whose measurable parameters do not fit exactly with the available ISs or there is a lack of suitable internal standards. The latter remark is associated with the fact that both quantitative and 2D structural analytical methods for mass spectrometry, so far, use mainly ISs. Thus, under the so-called confirmed structure, there is understood currently (a) a reported exact mass; (b) an unequivocally determined molecular formula, and (c) a single confirmed structure, which is obtained by means of IS. However, often, environmental and biological samples contain analytes lacking suitable ISs. Therefore, even 2D structural MS analysis produces a so-called possible structure or tentative candidates of a 2D chemical diagram. On the other hand, we should distinguish between so-called 2D chemical diagrams and 3D molecular structures as well. The 2D diagrams are obtained according to the rule of the degree of unsaturation, in addition to concepts of atomic valence and oxidation states. We note the following statements: (a) “…the sum of the valences of all the bonds formed by an ion is equal to the valence of the ion”, and (b) “…the stoichiometry must be obeyed by electro neutrality principle” [106]. However, 2D structures or 2D diagrams do not tell us anything about the chemical reactivity and the chemistry of the molecules. Why? The term “molecular structure” means a generic property determined by an ensemble of atoms in a molecule [106]. However, analytical statements, claiming 3D molecular structures, should be based on electronic structural analysis, which is reliable only when there is information about the electron density maps of the ensemble of atoms in the molecules. The electron density maps are proof of the probability density distribution, which is observable experimentally. These maps determine the probability of determining electrons at infinitely small volumes and positions in the 3D space [107]. The so-called total energy is determined on the basis of the electron density maps. Therefore, any 3D molecular model or 3D molecular structure is characterized by a unique to total energy quantity. In other words, from the perspective of structural chemistry, under a 3D molecular structure, there is understood a 3D molecular conformation and corresponding electronic structures, which are unique as a whole. There is a lack of corresponding disordered structural fragments.

Of course, an objection to the latter statements could be made by arguing that this study deals with the complicated case of the molecular structures of biocides showing a set of 3D molecular conformations, thus leading to a significant variation in *m*/*z* measurands and corresponding fluctuations in the observable *m*/*z* and intensity parameters of fragmentation ions. However, a part of the answer to such a question, if any, lies in the fact that the statements outlined above are obvious looking at the results from work [63] and those reported herein of the analysis of biocides in biota and sewage sludge.

Perhaps the most astonishing empirical evidence for the latter statements has been provided looking at the results from this study, analyzing the temporal distribution of measurable variables of standard PARA and its d_3_-derivative, particularly examining the fragmentation reactions of MS molecular ions [M+H]^+^ and [d_3_-M+H]^+^ at *m*/*z* 152, 155, 158, 174, 301, and 325, yielding the exact coefficient of linear correlation between D_QC_ and *D″*_SD_ data on CE = 25 V (Figure 7).

## 4. Materials and Methods

### 4.1. Chemicals and analytical instrumentation

Paracetamol (acetaminophen, 4′-hydroxyacetanilide, N-(4-hydroxy-phenyl)-acetamide), atenolol (2-[4-(2-hydroxy-3-isopropylamino-propoxy)-phenyl]-acetamide), propranolol (1-isopropylamino-3-(naphthalen-1-yloxy)-propan-2-ol), benzyl-dodecyl-dimethyl-ammonium chloride (BAC-C12), benzyl-dimethyl-tetradecyl-ammonium chloride (BAC-C14), benzyl-hexadecyl-dimethyl-ammonium chloride (BAC-C16), and benzyl-dimethyl-octadecyl-ammonium chloride (BAC-C18) were Sigma Aldrich products.

Thermo Finnigan LC (Massachusetts, USA) instrumentation equipped with a Micro AS autosampler and MSPump Plus was used. LC columns, namely the Waters Xbridge C18 column (Milford, USA; 1.0 × 100 mm ID, 3.5 μm), Waters Xselect charged surface hybrid C18 column (2.1 × 150 mm ID, 3.5 μm), Waters Xselect high-strength silica T3 column (1.0 × 100 mm ID, 3.5 μm), and a Phenomenex KrudKatcher Ultra 0.5 micron in-line filter, were used [1]. Experimental conditions of MS measurements are listed in Appendix A.

The study used the MS database on MS measurements available in [101,102].

### 4.2. Sample Preparation Methods, Samples, and Solutions

See details in [1,5,6,7,101,102,108].

### 4.3. Theory/Computations

The GAUSSIAN 98, 09; Dalton2011, and Gamess-US [109,110,111] program packages were employed. Ab initio and DFT molecular optimization was carried out by means of B3LYP, B3PW91, and ωB97X-D methods. Truhlar’s functional M06-2X was used [112]. The algorithm by Bernys was used to determine GSs. The stationary points at PES were obtained by harmonic vibrational analysis. The basis set cc-pVDZ by Dunning and 6-31++G(2d,2p) and quasirelativistic effective core pseudo-potentials from Stuttgart–Dresden(–Bonn) were used. MD computations were performed by ab initio BOMD, which was carried out with the M062X functional and SDD or cc-pvDZ basis sets, as well as without considering the periodic boundary condition. Allinger’s MM2 force field was utilized [113,114]. The low-order torsion terms were accounted for with higher priority than van der Waals interactions. The accuracy of the method compared with experiments was 1.5 kJ.mol^−1^ of diamante or 5.71.10^−4^ a.u.

### 4.4. Chemometrics

The software R4Cal 4.1.14 Open Office STATISTICs for Windows 7 was used. The statistical significance was checked by a *t*-test. The model fit was determined by an F-test. ANOVA was also used [115,116,117,118,119,120]. The ProteoWizard 3.0.11565.0 (2017), mMass 5.0.0, QuanBrowser 2.0.7 (Thermo Fischer Scientific Inc. Massachusetts, USA), and AMDIS 2.71 (2012) software were utilized.

## 5. Conclusions

Results from the study provide empirical evidence for the following conclusions.

(A)In testing the capability of Equation (2) [
$D_{SD}^{\prime \prime ,tot}=\sum_{i}^{n}D_{SD}^{\prime \prime ,i}=\sum_{i}^{n}2.6388.10^{-17}\times \bar{I_{i}^{2}}-{\left(\bar{I_{i}}\right)}^{2}$
] to quantify the MS intensity of analyte ions with a short span of scan time, we contrasted the use of classical quantitative methods based on ICIS and trapezoidal integration algorithms of peak detection. The analysis of surfactants in biota via equation ln[D″_SD_] = f(conc.) yields |r| = 0.9999_1_ examining the peaks of BAC-C12 at m/z 212.209 ± 0.1 and 211.75 ± 0.15.(B)Equation (4) [
$\bar{I^{TOT,q}}=\frac{1}{2}\times \frac{A_{I}^{q}}{A_{D}^{q}}\times D_{SD}^{\prime \prime ,tot}$
] has been proven for PARA ions. The relation between <I>_exp_ and <I>_theor_ shows|r|= 0.9942_9_ and 0.96501.(C)Parameter |r| = 1 has been obtained, determining the 3D molecular structures of PARA and its ions at m/z 152, 158, 174, 301, and 325 via the assessment of relation D″_SD_ = f(D_QC_) in biota at CE = 25 V.

## Figures and Tables

**Figure 1 ijms-24-06306-f001:**
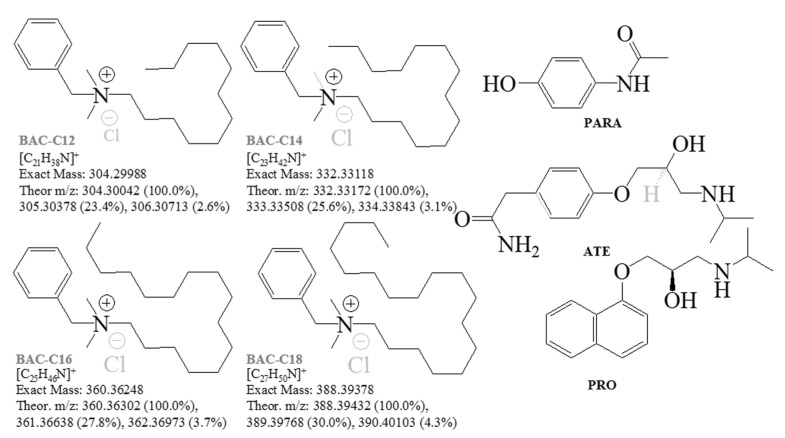
Chemical diagrams of benzalkonium chlorides and pharmaceutics; exact masses’ theoretical *m*/*z* data in cationic form [M]^+^; theoretical mass spectrometric isotopic compositions and intensity ratios of the isotope shapes.

**Figure 2 ijms-24-06306-f002:**
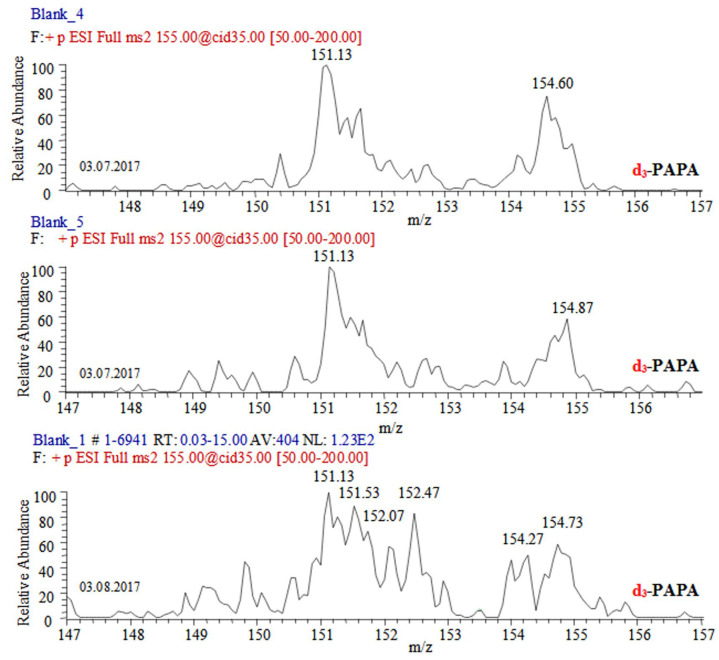
CID-MS/MS spectra of molecular cation [M+H]^+^ of d_3_-deuterated paracetamol at *m*/*z* 155 assessing within-day and between-day variability of mass spectrometric measurands.

**Figure 3 ijms-24-06306-f003:**
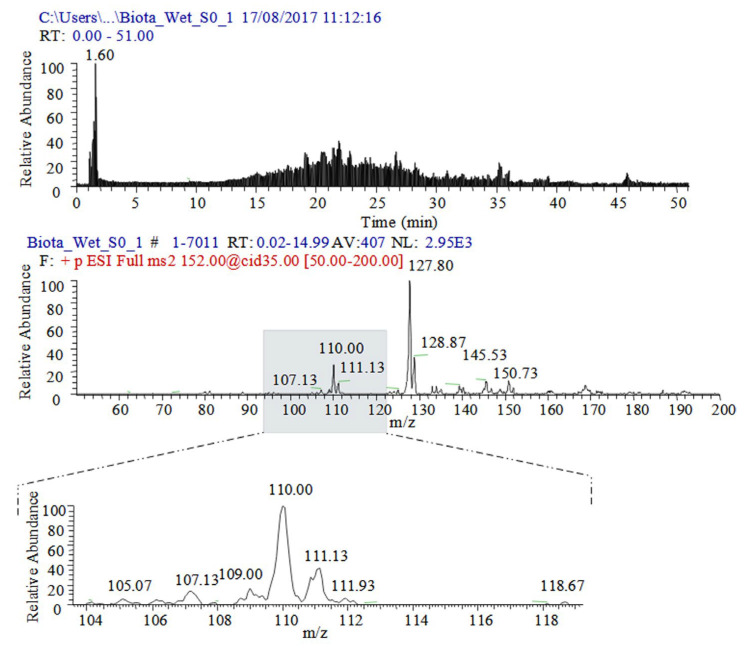
The total ion current (**top**) of standard sample of wet biota; CID-MS/MS spectrum of molecular cation [M+H]^+^ at *m*/*z* 152 of paracetamol (**bottom**).

**Figure 4 ijms-24-06306-f004:**
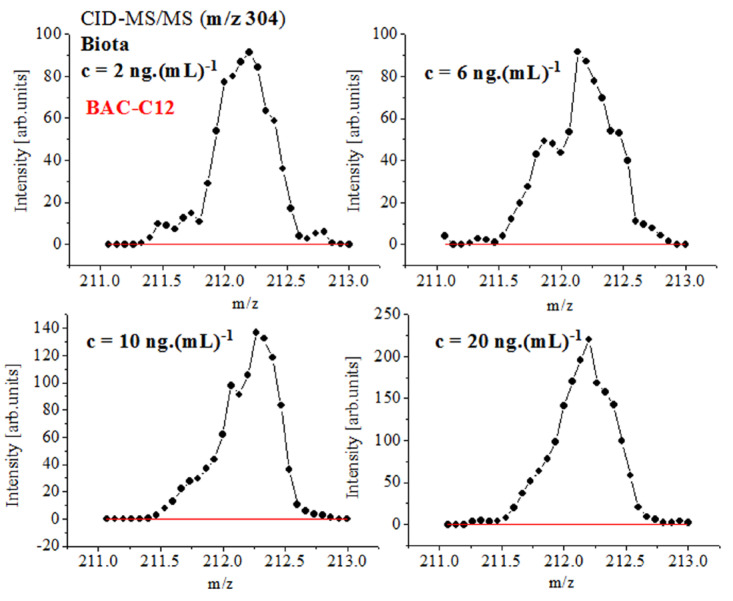
SRM spectra of analyte BAC-C12 in biota, using analyte molecular cation [M]^+^ at *m*/*z* 304 at concentrations c = 2, 6, 10, and 20 ng.(mL)^–1^; baseline correction of isotopologies; data on trapezoidal integration approach (see also Appendix A).

**Figure 5 ijms-24-06306-f005:**
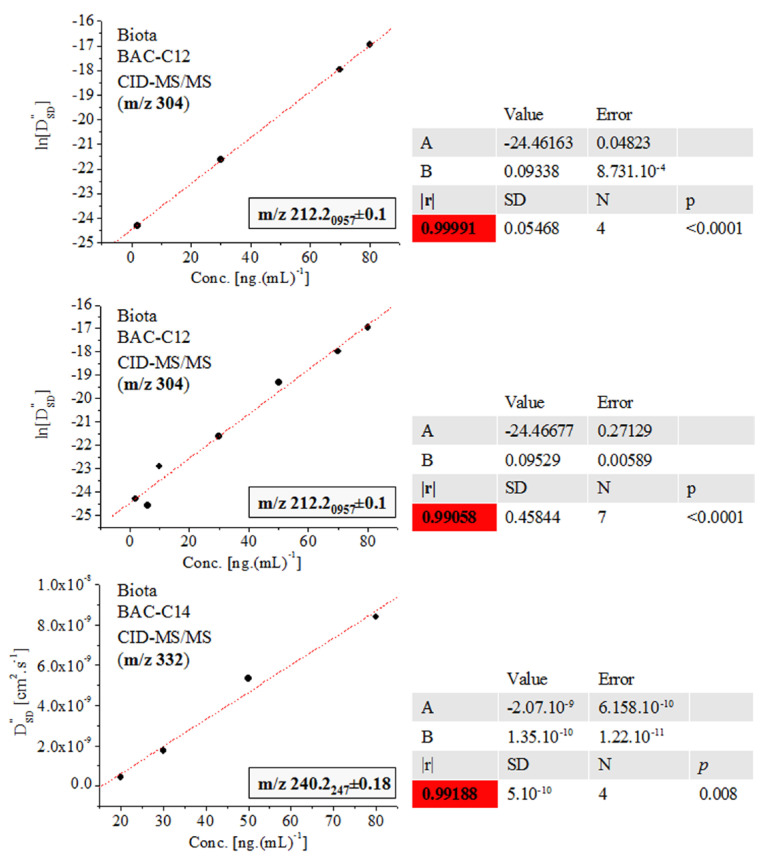
Linear functional relations ln[*D″*_SD_] = f(conc.) and *D″*_SD_ = f(conc.) of fragmentation ions at *m*/*z* 212.2_096_ and 240.2_247_ of SRM spectra of molecular cation [M]^+^ at *m*/*z* 304 and 332 of surfactants BAC-C12 and BAC-C14 in biota; chemometrics; the coefficient of correlation is highlighted in red.

**Figure 6 ijms-24-06306-f006:**
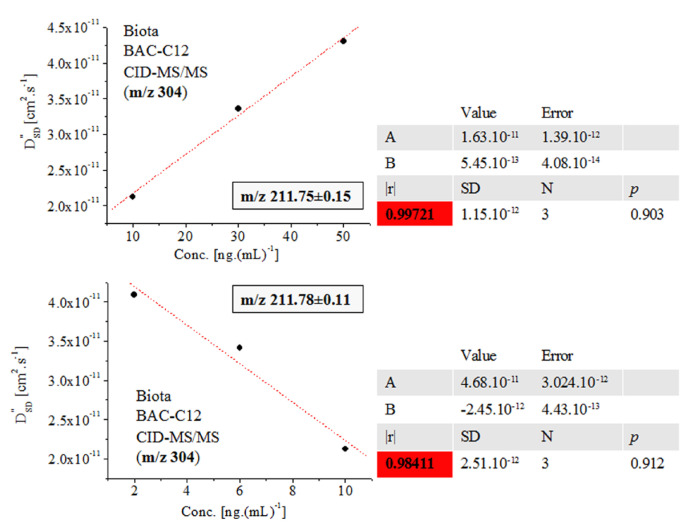
Linear functional relation *D″*_SD_ = f(conc.) of fragmentation ions at *m*/*z* 211.75 and 211.78 of SRM spectra of molecular cation [M]^+^ at *m*/*z* 304 of surfactant BAC-C12 in biota; chemometrics.

**Figure 7 ijms-24-06306-f007:**
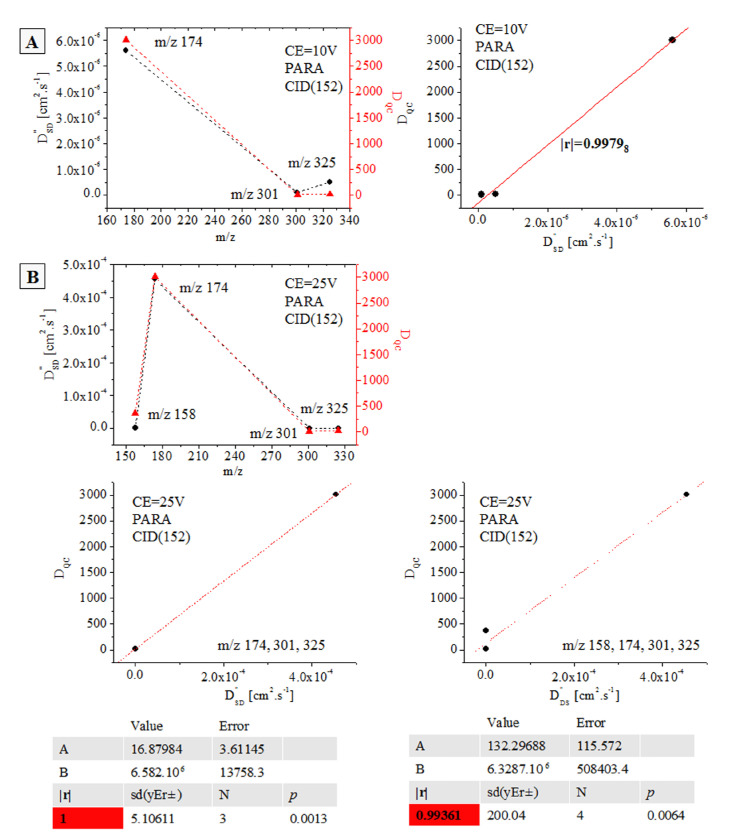
Functional relationships between *D″*_SD_ [cm^2^.s^−1^] and D_QC_ data on Equations (2) and (3) of fragmentation ions of CID-MS/MS reactions of molecular ion [M+H]^+^ of paracetamol at CE 10 (**A**) and 25 (**B**) V; chemometrics.

**Table 1 ijms-24-06306-t001:** Theoretical D_QC_ parameters of Equation (3) according to most stable analyte 3D molecular and electronic structures of species at ground and transition states and energetics (Appendix A); frequencies and atomic coordinates are listed in Appendix A.

m/z	Form	D_QC_	m/z	Form	D_QC_
110	110	2689.768			
152	152__a_	101.531209	155	155__a_	98.1605
	152__b_	646,318		155__b_	3.45123
158	158	367.4416	161	161	2902.9217
174	174	3009.373	177	177	42,352.98
325	325	21.08666	301	301	14.141939

## Data Availability

The experimental mass-spectrometric raw dataset can be download free of charge [101,102] at [https://doi.org/10.5281/zenodo.6543678]; [https://doi.org/10.5281/zenodo.6545447].

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
