# Peer review of "Stochastic Dynamic Mass Spectrometric Quantitative and Structural Analyses of Pharmaceutics and Biocides in Biota and Sewage Sludge"

_ijms, 2023, doi:10.3390/ijms24076306_

Round 1
Reviewer 1 Report
The manuscript is well written and contains enough novelty to be published in IJMS after a minor revision.
My comments/remarks:
· The manuscript is too long. Even the Introduction is over 6 pages. Nobody will read a 40 page paper. If You want to highlight your results it is better to summarize them in a max 20 page article.
· The English of the manuscript needs to be improved.
· Line 428 „kept cool at T =–20oC in an icebox. Then, specimens have been thawed at T = –26oC.” Please check the tempterature values!
· The quality of the figures especially Fig11 could have been better! Instead of screen captures, use processed images next time!
· Table 1 should be placed in the supporting information.
· In Figure 15 the energy values are not in arbitrary units. I think they should be given in Hartree which can be converted to kJ.
Author Response
The manuscript is too long. Even the Introduction is over 6 pages. Nobody will read a 40 page paper. If You want to highlight your results it is better to summarize them in a max 20 page article.
The manuscript is reduced to 21 pages owing to large figures.
The English of the manuscript needs to be improved.
The English is corrected, accordingly.
Line 428 „kept cool at T =–20oC in an icebox. Then, specimens have been thawed at T = –26oC.” Please check the tempterature values!
The sub-section is new.
The quality of the figures especially Fig11 could have been better! Instead of screen captures, use processed images next time!
The quality of the figures is improved increasing in dpi. The old Figure 11 (Figure 4 in the revised text is completely new.)
Table 1 should be placed in the supporting information.
Table 1 is moved to supporting information file. It is shown as Table A8.
In Figure 15 the energy values are not in arbitrary units. I think they should be given in Hartree which can be converted to kJ.
The units are astronomical units or Hartree as they are authomatically generated by computations. It is corrected in the revised version oft he text. Please, consider Figure A31.
Reviewer 2 Report
Dear Authors,
The presented manuscript and scientific material are very interesting so I propose to publish it in IJMS after some small corrections.
The text needs some corrections
Line 35. Why sentence is in brackets. (… .). The text should be checked and some minor language/editorial mistakes should be corrected.
Chemical structures of the studied compounds should be added into the Introduction.
The broad Introduction is interesting. However, the question is if the text was used before in some PhD Thesis? The Introduction should be checked for plagiarism.
Best regards
Author Response
The presented manuscript and scientific material are very interesting so I propose to publish it in IJMS after some small corrections.
The text needs some corrections
Line 35. Why sentence is in brackets. (… .). The text should be checked and some minor language/editorial mistakes should be corrected.
The text is completely corrected and revised, moreover, the introductory section is reduced, significantly. The employment in brackets is usually when the sentence is not of so significantly important, but would be useful and informative in the context of the corresponding paragraph.
Chemical structures of the studied compounds should be added into the Introduction.
The chemical diagrams of the analytes are shown as new Figure 1 in the introductory section.
The broad Introduction is interesting. However, the question is if the text was used before in some PhD Thesis? The Introduction should be checked for plagiarism.
Declaration: I am author of the whole content of the old and the revised version of the text, including the introductory section. Also, I am author of the figures, as well. There is a lack of plagiarism.
The declaration above does not restrict the text of the study to be object of independent examination for plagiarism.